# Lost in Aggregation: How Benchmarks Overlook Irreplaceable Model Strengths

## Abstract

Tabular machine learning benchmarks typically summarize performance by averaging scores, ranks, or pairwise wins across datasets. Such aggregates are useful for selecting robust default models, but they can obscure a different question: which models are necessary to attain peak performance on particular datasets? We argue that benchmark evaluation should also consider the data-centric peak performance frontier, defined by the best statistically supported performance achieved on each dataset. From this perspective, a model may be irreplaceable, sufficient, redundant, or fallible depending on where it lies on the frontier relative to other models. Applying this framework to the TabArena benchmark, we find that common aggregation metrics are highly correlated and largely measure consistency and avoiding failures, while being much less aligned with dataset-level irreplaceability. Consequently, models performing decently across datasets without ever being the best choice are rewarded while models with unique dataset-specific strengths appear mediocre under aggregation. Hence, benchmark progress should be measured not only by improvements on aggregation metrics but also by whether new models expand the set of attainable peak performances across datasets.

## 1. Introduction

Benchmarking is a central mechanism for measuring progress in machine learning (Vanschoren et al., 2014; Erickson et al., 2025). By evaluating many models on a variety of datasets, benchmarks provide common reference points for comparing methods, identifying strong baselines, and guiding model development. In tabular machine learning, this role is especially important because datasets vary widely

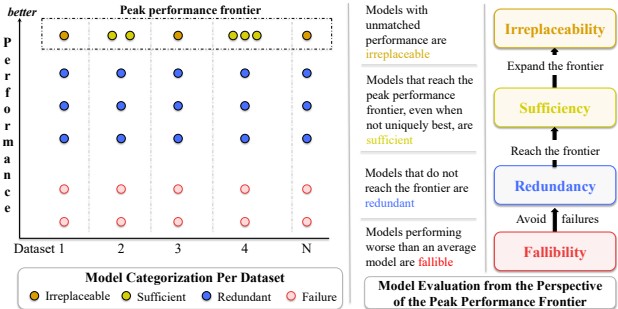

*Figure 1.* **Visualization of the data-centric peak performance frontier and the disentangled aggregation metric dimensions.**

in sample size, feature types, noise levels, and interaction patterns (Grinsztajn et al., 2022; Tschalzev et al., 2024). To make large-scale comparisons tractable, benchmark results are usually compressed into aggregate scores such as average ranks, win rates, normalized errors, or Elo ratings (Demšar, 2006; Liu et al., 2024; Erickson et al., 2025). These aggregates are useful when the goal is to identify a robust default model. However, they conflate qualitatively different evaluation dimensions.

We therefore argue for evaluating models relative to the data-centric peak performance frontier (Figure 1): the frontier spanned by the best statistically supported performance on each dataset. From this perspective, the relevant unit of analysis is not only the model's average position across tasks, but also its role on each individual dataset [1]. Accordingly, whenever a set of candidate methods is evaluated on a new dataset, the performance of each method can be categorized into one out of four practically distinct conclusions about the performance of each model: (**1**) it achieves the best performance and is unmatched, hence the model is irreplaceable for peak performance on this dataset. (**2**) it achieves high performance that is indistinguishable from at least one other model, hence the model is a sufficient solution for this dataset, although alternatives exist. (**3**) it does not reach the frontier, but also does not fail decisively, hence it is redundant on this dataset. (**4**) it performs substantially worse than most other models, hence it is fallible on this dataset. In this

---

[1]Anonymous Institution, Anonymous City, Anonymous Region, Anonymous Country. Correspondence to: Anonymous Author <anon.email@domain.com>.

Preliminary work. Under review by the International Conference on Machine Learning (ICML). Do not distribute.

---

[1]Note that in this paper we focus on a comparison of the predictive performance of models, leaving other aspects like efficiency, hardware constraints, or ensembling aside.

paper, we study how well common benchmark aggregation metrics capture this frontier-oriented view.

## 2. Disentangling Benchmark Performance

To define performance dimensions relative to the data-centric peak performance frontier, we require three ingredients: (1) a notion of peak performance, (2) a criterion for statistically indistinguishable performance, and (3) a criterion for decisive failure.

**Indistinguishable Performance**  Benchmark protocols typically rely on repeated measurements through cross-validation or repeated train-test splits (Dietterich, 1998; Bouckaert & Frank, 2004). TabArena (Erickson et al., 2025), for example, evaluates models using repeated 3-fold cross-validation, with 10 repetitions for small datasets and 9 repetitions for medium-sized datasets. In the benchmark evaluation, these repetitions are mainly used to stabilize aggregate estimates, although more generally they can be viewed as a way to quantify uncertainty at the level of individual datasets.

For each dataset $d$, we observe model performance over matched evaluation folds. Let $s_m^{(r)}$ denote the error of model $m$ on fold $r = 1, \ldots, R_d$, where lower values indicate better performance. To decide whether a candidate model $m_i$ significantly outperforms a reference model $m_j$ on dataset $d$, we use the paired fold-wise differences $\delta^{(r)} = s_{m_i}^{(r)} - s_{m_j}^{(r)}$, and apply a one-sided Wilcoxon signed-rank test with $\alpha = 0.05$ (Wilcoxon, 1945) to the paired differences. Since lower error is better, $m_i$ is considered significantly better than $m_j$ if the test rejects the null of no improvement in favor of the alternative that the paired differences are shifted below zero, at significance level $\alpha$. This complements classical benchmark statistics, which usually collapse this information before aggregation (Demšar, 2006; Garcia & Herrera, 2008; Gijsbers et al., 2019), thus treating small and statistically unsupported differences similarly to reliable ones.

**Peak Performance.**  We define peak performance relative to the empirical best model on each dataset. For dataset $d$, let $m_d^\star$ denote the model with the lowest average error across folds. The peak-performance set is

$$\mathcal{P}_d = \{m \in \mathcal{M} : m_d^\star \text{ is not significantly better than } m\}.$$

Thus, $\mathcal{P}_d$ contains all models that are not statistically separated from the empirical best model under the test described above. Models outside $\mathcal{P}_d$ are significantly below peak performance on dataset $d$.

A model is *irreplaceable* on dataset $d$ if it is the only member of $\mathcal{P}_d$. The irreplaceability of a model $m$ is the number of datasets on which it is the sole peak-performing model:

$$\text{Irreplaceability}(m) = |\{d \in \mathcal{D} \ : \ \mathcal{P}_d = \{m\}\}| .$$

A nonzero irreplaceability score indicates that the model exhibits unique strengths on specific datasets and contributes non-redundant value to the peak-performance frontier.

The sufficiency of a model $m$ is the number of datasets on which it belongs to the peak-performance set:

$$\text{Sufficiency}(m) = |\{d \in \mathcal{D} \ : \ m \in \mathcal{P}_d\}| .$$

A high sufficiency score indicates that the model frequently reaches the peak-performance frontier and can therefore often serve as a sufficient stand-alone choice, even when other models reach the same level as well.

**Falling Behind.**  We define "falling behind" on a dataset as being significantly worse than an average model. For each dataset $d$, we define $m_d^{\text{med}}$ as the model with median average performance across folds. The fallibility of a model $m$ is then the number of datasets on which the median model is significantly better than $m$:

$$\text{Fallibility}(m) = \left|\left\{d \in \mathcal{D} \ : \ m_d^{\text{med}} >> m_d\right\}\right| .$$

A high fallibility score indicates that a model is not merely below peak performance, but significantly worse than a typical benchmark competitor.

Models that are neither sufficient nor fallible occupy the remaining category. We call such models *redundant*: they do not reach the peak-performance frontier, but they are also not significantly worse than the median-performing model. The redundancy category can be further refined, for example by counting how often a model is statistically indistinguishable from the second-, third-, or lower-ranked model.

## 3. Analysis on the TabArena Benchmark

**Experimental Setup**  For our analysis we use TabArena (Erickson et al., 2025), which is a maintained benchmark with a live leaderboard and public, extensive evaluation results. It improves over the evaluation of prior benchmarks (McElfresh et al., 2023; Grinsztajn et al., 2022; Tschalzev et al., 2025) using a stricter data curation and more extensive evaluation protocol, and currently comprises 27 models evaluated on 51 carefully curated datasets. We compare our dataset-centric metrics against the leaderboard metrics. At the time of writing, the live leaderboard shows five performance aggregation metrics: ELO, normalized error (score), rank, harmonic mean rank, and improvability. ELO is used as the main metric, and improvability as the secondary metric. We provide more details and snapshots of the current benchmark results in Appendix A. For all experiments we use $\alpha = 0.05$ for significance tests.

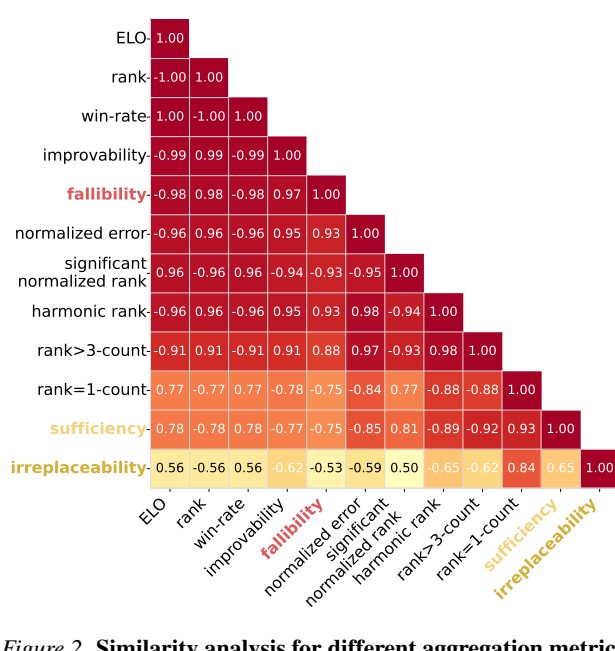

*Figure 2.* **Similarity analysis for different aggregation metrics.** We show the pairwise correlation heatmap ordered by absolute correlation to ELO with ELO shown first. To focus on peak performance, correlations are computed on the set of models that ranks in the top 10 on at least one of the two major metrics in TabArena, and ours: ELO, improvability, irreplaceability, sufficiency.

### 3.1. Benchmarks Favor Consistency & Avoiding Failure

We first ask how closely the proposed frontier-oriented metrics align with the aggregation metrics reported by the TabArena leaderboard.

**Aggregate metrics largely measure the same dimension.** Figure 2 shows that the standard aggregation metrics are highly correlated across models. Elo, average rank, and win rate induce nearly identical model orderings; normalized error, harmonic mean rank, and improvability are also largely aligned with them. Thus, although these metrics are mathematically different, they largely summarize the same dominant performance dimension. In contrast, irreplaceability and sufficiency are substantially less correlated with the standard metrics, suggesting that they capture a different notion of model value: unique contributions to the dataset-wise peak-performance frontier.

**Aggregation rewards consistency.** To understand what this dominant aggregation dimension rewards, we add a hypothetical model whose metric error rank is fixed on every fold of every dataset and recompute the leaderboard metrics while varying this fixed rank from last to first. As shown in Figure 3 (left), a model can reach the top of several leaderboard metrics without ever attaining dataset-level peak performance. For example, a model ranked sixth on every dataset can still rank first under Elo and average rank. Thus, aggregation can reward consistently decent perfor-

mance over dataset-level excellence. This pattern appears, to varying degrees, across all TabArena leaderboard metrics. Harmonic mean rank is more aligned with the peak-performance frontier, but, nevertheless, none of these metrics directly asks whether a model is uniquely best on any dataset, which motivates the introduced dataset-centric frontier metrics.

**Aggregation rewards failure avoidance.** The correlation analysis already suggests that standard leaderboard metrics are more sensitive to failure avoidance than to unique dataset-level excellence: among our proposed dimensions, fallibility is most strongly aligned with the standard metrics, whereas irreplaceability is the least aligned. To test this interpretation more directly, we add a hypothetical model that ranks either first or last on every fold of each dataset, and recompute the leaderboard metrics while varying the fraction of datasets on which the hypothetical model ranks first. The resulting progression in Figure 3 (right) shows that even when the model is first on more than half of all datasets, it does not even reach the top 10 under four of the five main metrics, with harmonic mean rank being the only exception.

### 3.2. Frontier Metrics Reveal Unique Strengths

Figure 4 summarizes the analysis from the data-centric peak performance perspective in one plot. Irreplaceability identifies datasets on which a model expands the peak-performance frontier and is statistically unmatched. Hence, nonzero irreplaceability suggests that a model has a distinctive inductive bias that is uniquely beneficial for at least one dataset. Overall, 9/27 models analyzed are irreplaceable, indicating unique inductive biases.

**Similar aggregate performance hides distinct strengths.** TabICLv2 and TabPFN-2.5 appear similar on standard leaderboard metrics, yet each is irreplaceable on several distinct datasets, showing that similarly ranked models are not necessarily substitutable. Therefore, using our metrics leads to meaningfully different conclusions than standard aggregate rankings, whenever models behave similar on average while having distinct strengths providing value in diagnostic interpretation.

**Strong aggregate performance ≠ irreplaceability.** Figure 4 shows that strong aggregate performance does not necessarily imply strong frontier coverage. For example, RealMLP (Holzmüller et al., 2024) ranked first at the time of the TabArena publication (Erickson et al., 2025), yet in our analysis it ranks 13th on sufficiency, is never irreplaceable and reaches the peak-performance frontier on only two datasets. Instead, its strength lies mainly in avoiding poor outcomes: it has the second-best fallibility score and the fourth-best count of ranks greater than or equal to three among all competitors.

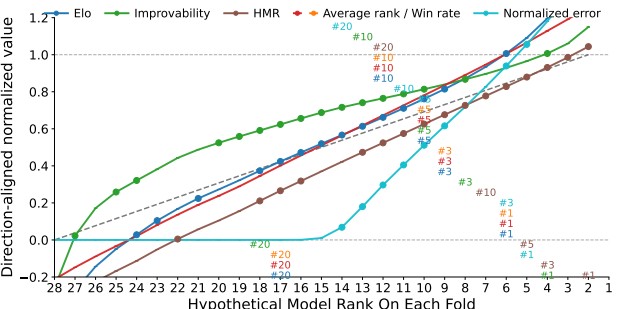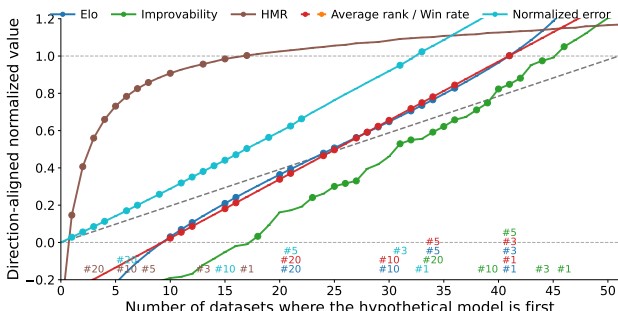

*Figure 3.* **Behavior of standard aggregation metrics under hypothetical model profiles.** Numbers inside the plot indicate a rank change on a metric, e.g., #1 means that the hypothetical model reaches rank 1 at that point in the progression. (Left) A hypothetical model is assigned the same rank on every fold of every dataset, moving from last to first. We show that under aggregation metrics, a consistently middling model can rank highly without ever reaching dataset-level peak performance. (Right) A hypothetical model ranks either first or last on each dataset, with the fraction of first-place datasets varied. We show that most aggregation metrics remain low even when the hypothetical model is irreplaceable on most datasets, indicating that these metrics penalize failures more strongly than they reward isolated frontier contributions.

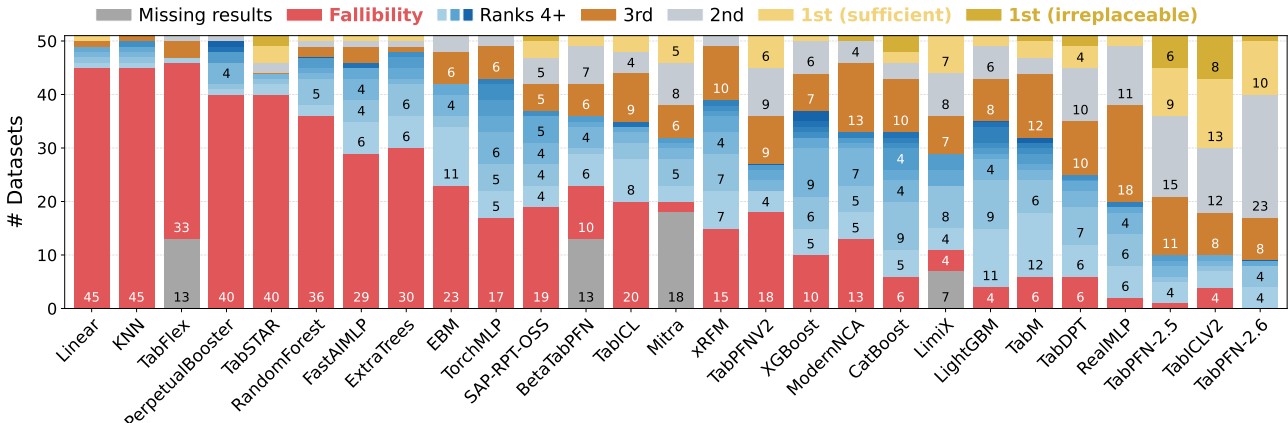

*Figure 4.* **Combined view of the model performance dimensions.** We show the significant rank counts for each model focusing on the introduced metrics. 1st (sufficient) illustrates sufficiency minus the no. of datasets on which a model is irreplaceable to make the bar plots add up to 51 datasets. The models are sorted by ELO (rightmost is best).

**Weak aggregate performance $\neq$ model irrelevance.** Some models, most notably CatBoost (Prokhorenkova et al., 2018), TabDPT (Ma et al., 2024) and ModernNCA (Ye et al., 2024) reach peak performance on particular datasets, where no other evaluated model reaches the peak performance frontier. At the same time the models fail on many datasets and rarely are sufficient. An important distinction is that, from an AutoML perspective, such a model may be undesirable, while from a model perspective, it can be highly useful in revealing inductive biases that are not captured by consistently strong models. Irreplaceability can therefore be used to evaluate individual models, while aggregation metrics are more suitable to evaluate AutoML solutions.

**Irreplaceability Can Reveal Pseudo-Improvements.** By copying the current best open source model and making random changes, a new model proposal can look like a strong

contribution without offering anything new. Using our irreplaceability metric can expose this by identifying whether a model provides unique value despite having aggregate performance similar to other high-performing models.

## 4. Conclusion

Our analysis suggests that current aggregation metrics well suited for identifying robust default models, but do not sufficiently reward models that expand the peak-performance frontier. By disentangling default utility from irreplaceability, sufficiency, and fallibility, our framework makes it easier to identify when a new model contributes non-redundant value to the benchmark model zoo. Future work can build on our evaluation and analyze inductive biases leading to irreplaceable model performance.

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

## A. Details on the TabArena Benchmark and its Aggregation Metrics

**TabArena summary.** TabArena (Erickson et al., 2025) is a continuously maintained benchmark for tabular machine learning whose main strength is that it is designed to remain up-to-date as datasets, model implementations, validation protocols, and leaderboard results evolve. Rather than providing a fixed one-time comparison, it combines manually curated datasets, well-implemented baselines and state-of-the-art models, reproducible evaluation code, stored predictions, and public maintenance protocols. The benchmark is initialized with a representative collection of practical IID tabular datasets spanning classification and regression tasks, and it evaluates a broad range of model families, including gradient-boosted tree methods, classical machine learning baselines, deep tabular learning models, foundation models for tabular data, hyperparameter-tuned variants, and ensembles. This breadth allows TabArena to compare not only individual models, but also the effects of validation strategy, tuning budget, and ensembling, making it useful for assessing practical tabular ML performance. At the time of writing, the benchmark offers open-sourced evaluation results for 27 tabular models on 51 datasets (see Table 1).

**Per-dataset and per-fold error.** For each model $m$, let $e_{m,r}$ be the error value of model $m$ on fold $r$ of dataset $d$. The task-specific error is

$$e_{m,r} = \begin{cases} 1 - \text{AUROC}_{m,r}, & \text{binary classification,} \\ \text{LogLoss}_{m,r}, & \text{multiclass classification,} \\ \text{RMSE}_{m,r}, & \text{regression.} \end{cases}$$

These errors are then aggregated over evaluation units using Elo, normalized score, average rank, harmonic mean rank, and pairwise win rates, and improvability.

**Aggregation units.** The aggregation metrics below can be computed over one of two evaluation units $\mathcal{U}$. Depending on the chosen setting, an evaluation unit $u \in \mathcal{U}$ is either a full dataset, after averaging over its outer folds $r$, or a single dataset–fold pair. Thus, all metrics operate on errors $e_{m,u}$, where $m$ denotes the model and $u$ denotes the evaluation unit. We compare the metrics with aggregation over datasets, which is the TabArena default and the statistically correct version, since aggregating over folds treats each fold as an independent dataset.

**Elo.** Elo is the primary aggregation metric used by TabArena. It treats model comparison as a collection of pairwise matches across evaluation units. For each evaluation unit $u$, two models $i$ and $j$ are compared using their errors $e_{i,u}$ and $e_{j,u}$: the model with lower error is counted as the winner. These pairwise outcomes are then aggregated into Elo ratings.

If $R_i$ and $R_j$ are the Elo ratings of models $i$ and $j$, then the expected win probability of model $i$ against model $j$ is

$$P(i \succ j) = \frac{1}{1 + 10^{(R_j - R_i)/400}}.$$

Thus, a 400-point Elo advantage corresponds to an expected win ratio of about $10 : 1$, or roughly a $91\%$ expected win probability. TabArena uses Elo as the main leaderboard metric because it gives each evaluation unit equal influence and is robust to differences in the scale of the underlying task metrics. However, Elo only depends on win/loss outcomes and therefore ignores the magnitude of performance differences. TabArena calibrates ELO such that 1000 ELO matches the performance of a default random forest.

**Normalized score.** For each evaluation unit $u$, errors are linearly rescaled so that the best method receives score 1 and the median method receives score 0. Scores below zero are clipped:

$$s_{m,u} = \max\left(0, \frac{e_{\text{median},u} - e_{m,u}}{e_{\text{median},u} - e_{\text{best},u}}\right),$$

where

$$e_{\text{best},u} = \min_m e_{m,u}.$$

The final normalized score is the average across evaluation units:

$$S_m = \frac{1}{|\mathcal{U}|} \sum_{u \in \mathcal{U}} s_{m,u}.$$

Higher is better. This metric preserves some information about the magnitude of error differences, unlike Elo or rank.

**Average rank.** For each evaluation unit, models are ranked by error, with lower error giving better rank. Let $r_{m,u}$ be the rank of model $m$ on evaluation unit $u$. The average rank is

$$\text{AvgRank}(m) = \frac{1}{|\mathcal{U}|} \sum_{u \in \mathcal{U}} r_{m,u}.$$

Lower is better. This metric is scale-free but discards the magnitude of performance differences.

**Harmonic mean rank.** The harmonic mean rank is

$$\text{HMeanRank}(m) = \frac{|\mathcal{U}|}{\sum_{u \in \mathcal{U}} \frac{1}{r_{m,u}}}.$$

Lower is better. Because harmonic means emphasize small ranks, this metric rewards models that are occasionally very strong, even if they are not consistently strong across all evaluation units.

**Improvability.** TabArena introduces improvability to measure how much better the best observed method is than a given method on each evaluation unit. For model $m$ and evaluation unit $u$:

$$\text{Imp}_{m,u} = 1 - \frac{e_{\text{best},u}}{e_{m,u}}, \qquad e_{\text{best},u} = \min_{m'} e_{m',u}.$$

The aggregate improvability is

$$\text{Imp}(m) = \frac{1}{|\mathcal{U}|} \sum_{u \in \mathcal{U}} \text{Imp}_{m,u}.$$

Lower is better; $\text{Imp}(m) = 0$ means the method matches the best observed method on every evaluation unit. Unlike Elo and rank, improvability is sensitive to the magnitude of performance gaps. Furthermore, this metric is implicitly designed to take the peak performance frontier into account, but as we show is still biased towards rewarding consistency and avoiding failures.

**Rank-1 count and rank-$> 3$ count.** In addition to average rank and harmonic mean rank, one can summarize ranks focusing on the top of the leaderboard. The rank-1 count measures how often a model is the best-performing method on an evaluation unit:

$$\text{Rank1Count}(m) = \sum_{u \in \mathcal{U}} \mathbf{1}[r_{m,u} = 1].$$

Higher is better. This metric is equivalent to counting the number of evaluation units on which the model achieves first place, up to the chosen tie-handling rule.

The rank-$> 3$ count measures how often a model falls outside the top three methods:

$$\text{RankGreater3Count}(m) = \sum_{u \in \mathcal{U}} \mathbf{1}[r_{m,u} > 3].$$

Lower is better. While rank-1 count attempts to highlight peak performance, rank-$> 3$ count captures how often a method fails to remain among the strongest competitors. Note that these metrics are not part of the leaderboard of the official TabArena website, but can also be computed using the codebase. In contrast to these metrics, we use significance tests to prevent insignificant results from biasing the rank counts.

*Table 1.* Evaluated models in TabArena. Family abbreviations: GBDT = gradient-boosted decision trees; RF/ET = random forest / extra trees; MLP = multilayer perceptron-style neural networks; TFM = tabular foundation model; ICL = in-context learning; RET = retrieval-based model; AutoML = automated machine learning system.

| Model | Evaluated configuration | Family |
|---|---|---|
| | | |

*Continued on next page*

| Model | Evaluated configuration | Family |
|---|---|---|
| Linear model (Erickson et al., 2020) | tuned + ensemble | Baseline |
| kNN (Erickson et al., 2020) | tuned + ensemble | Baseline |
| Extra Trees (Geurts et al., 2006) | tuned + ensemble | Baseline |
| Random Forest (Breiman, 2001) | tuned + ensemble | Baseline |
| Explainable Boosting Machine (Lou et al., 2013) | tuned + ensemble | Baseline |
| XGBoost (Chen & Guestrin, 2016) | tuned + ensemble | GBDT |
| LightGBM (Ke et al., 2017) | tuned + ensemble | GBDT |
| CatBoost (Prokhorenkova et al., 2018) | tuned + ensemble | GBDT |
| PerpetualBooster (https://github.com/perpetual-ml/perpetual) | tuned + ensemble | GBDT |
| FastAIMLP (Erickson et al., 2020) | tuned + ensemble | MLP |
| TorchMLP (Erickson et al., 2020) | tuned + ensemble | MLP |
| RealMLP (Holzmüller et al., 2024) | tuned + ensemble | MLP |
| ModernNCA (Ye et al., 2024) | tuned + ensemble | MLP |
| TabM (Gorishniy et al., 2024) | tuned + ensemble | MLP |
| xRFM (Beaglehole et al., 2025) | tuned + ensemble | Other |
| TabDPT (Ma et al., 2024) | tuned + ensemble | TFM |
| TabFlex (Zeng et al., 2024) | default | TFM |
| TabICL (Qu et al., 2025) | default | TFM |
| TabPFNv2 (Hollmann et al., 2025) | tuned + ensemble | TFM |
| Mitra (Zhang et al., 2025a) | default | TFM |
| Limix (Zhang et al., 2025b) | default | TFM |
| SAP-RPT-OSS (Spinaci et al., 2025) | default | TFM |
| BetaTabPFN (Liu & Ye, 2025) | default | TFM |
| TabSTAR (Arazi et al., 2025) | tuned + ensemble | TFM |
| TabPFN-2.5 (Grinsztajn et al., 2025) | tuned + ensemble | TFM |
| TabICLv2 (Qu et al., 2026) | default | TFM |
| TabPFN-2.6 (Prior Labs, 2025) | default | TFM |
| AutoGluon (Erickson et al., 2020) | 1.4 (best), 1.4 (extreme), 1.5 (extreme) | AutoML |

## B. Intended Use of the Peak Performance Frontier Metrics

The metrics introduced in section 2 are intended as complementary descriptors of benchmark performance. They are not intended as a replacement for aggregate leaderboard scores. Standard aggregation metrics remain useful for identifying strong default models, especially when the goal is robust performance across many datasets. Peak performance frontier metrics instead disentangle this aggregate behavior into more interpretable and actionable dataset-level properties: whether a model uniquely expands the peak-performance frontier, frequently reaches it, remains competitive without reaching it, or fails decisively.

**Interpreting the metrics.** We interpret the performance dimensions as follows:

1. Irreplaceability is the strongest indicator of model development progress. It counts datasets on which a model is uniquely necessary to achieve statistically supported peak performance. It is therefore the main dimension for assessing whether a new model expands the state of the art or adds novel capabilities to the existing model zoo.

2. Sufficiency captures practical frontier coverage. A model with high sufficiency frequently matches the best available performance, even when alternatives exist. It is therefore useful for evaluating models as strong standalone choices or default starting points for modeling.

3. Fallibility captures decisive failure. Low fallibility is important for automation, prototyping, and deployment, because it indicates that a model rarely performs significantly worse than a typical benchmark competitor.

4. Redundancy captures cases where a model does not reach the peak-performance frontier but also does not fail decisively. We intentionally keep this category broad: within it, finer distinctions can be made, for example by asking whether a model is statistically indistinguishable from the second-, third-, or lower-ranked model. However, for the present analysis, we group these cases together because our main focus is whether a model contributes to peak performance.

5. **Default Utility** summarizes the information captured by standard aggregate metrics such as Elo, average rank, win rate, normalized error, and improvability. This dimension reflects how reliably a model performs across the full benchmark and is useful for selecting robust default methods. Unlike irreplaceability or sufficiency, however, it does not directly indicate whether a model expands or reaches the peak-performance frontier on individual datasets.

These dimensions support different notions of improvement. Moving datasets from fallibility to redundancy improves robustness, but does not establish peak performance. Moving from redundancy to sufficiency indicates practical usefulness, because the model becomes a valid choice for reaching the frontier. Moving from sufficiency to irreplaceability marks the most convincing form of benchmark progress because the model becomes uniquely necessary for achieving the best known performance on at least one dataset.

**Model development view.** The frontier view separates two objectives that are often conflated by aggregate metrics. The first is default-model quality: whether a method performs consistently well and avoids poor outcomes. This objective is central for AutoML, prototyping, and deployment, where low fallibility and high sufficiency are desirable. The second is frontier contribution: whether a method achieves peak performance that other models do not match. This objective is central for benchmark-driven model development, where irreplaceability can reveal dataset regimes in which a model has a distinctive inductive bias. Thus, a model may be valuable even if it is not a good default choice, and a strong default model may still contribute little unique value to the peak-performance frontier.

**Practitioner view.** For a practitioner seeking strong performance on a new dataset, looking at Figure 4 can give a hint on which models could be tried in which order using a staged evaluation strategy: First, train a model with low fallibility and general robustness for reliable initial results. Second, try a model with high sufficiency for a high empirical probability of reaching the peak-performance frontier. Third, evaluate irreplaceable models ordered by their irreplaceability scores to test whether distinct inductive biases are favorable for the given dataset. Based on the TabArena models, this corresponds to starting with a robust high-performing default (e.g. TabPFN2.6), then testing the models with the highest frontier (e.g., TabICLv2), and finally considering irreplaceable models such as TabPFN-2.5, CatBoost, TabDPT, TabSTAR[2], TabM, ModernNCA, and SAP-RPT-OSS. Note that in practice, the workflow will also depend on computational budget, deployment constraints, and domain knowledge, but the general principle is to start with robust defaults followed by models that provide non-redundant frontier coverage and irreplaceable strengths.

**Benchmarking view.** From the benchmarking perspective, Figure 4 can be used to compactly summarize the peak performance view in addition to a plot summarizing the dimensions measured by currently dominant aggregation metrics. Importantly, a method can only be irreplaceable if it is uniquely the best. Hence, one has to be careful with near-identical clones of open source models as well as future versions of the same model being evaluated. To better guard against that, we present an alternative in Figure 9, where for datasets on which multiple models are sufficient, the models which first achieved irreplaceability over previously existing models are highlighted. This rewards models for expanding the peak performance frontier in comparison to later models which "just" match this performance.

## C. Additional Results

In this section, we provide additional evaluation results using the proposed metrics as well as ablations on the proposed metrics.

**Sensitivity to the significance level** Our framework depends on the utilized dataset-level significance test and the chosen significance level. Since we chose somewhat suggestive wording for the metric names, it is essential to analyze alternative versions of the results with varying definitions of significant performance differences before settling on a significance testing strategy. Figure 5 shows that for the Wilcoxon signed-rank test that we employ, the significance level mainly determines on how many datasets differences are considered insignificant. In practice, significance levels above 0.1 are rarely used, and $\{0.01, 0.05, 0.1\}$ are the most common conventions. Figure 5 shows that these common significance levels offer good signal-to-noise trade-offs for the Wilcoxon signed-rank test, while for higher significance levels noise is introduced with more models included that show generally worse performance also under common aggregation.

**Irreplaceability of tuning** In Figure 6, we study whether hyperparameter tuning and/or post-hoc ensembling of configura-

---

[2]Note that TabSTAR's strong performance may result from leaks since part of the TabArena datasets were used for pretraining the model. Our framework implicitly assumes that such issues are not present in the benchmark. Hence, unusual results on our metrics can also help to identify data leaks.

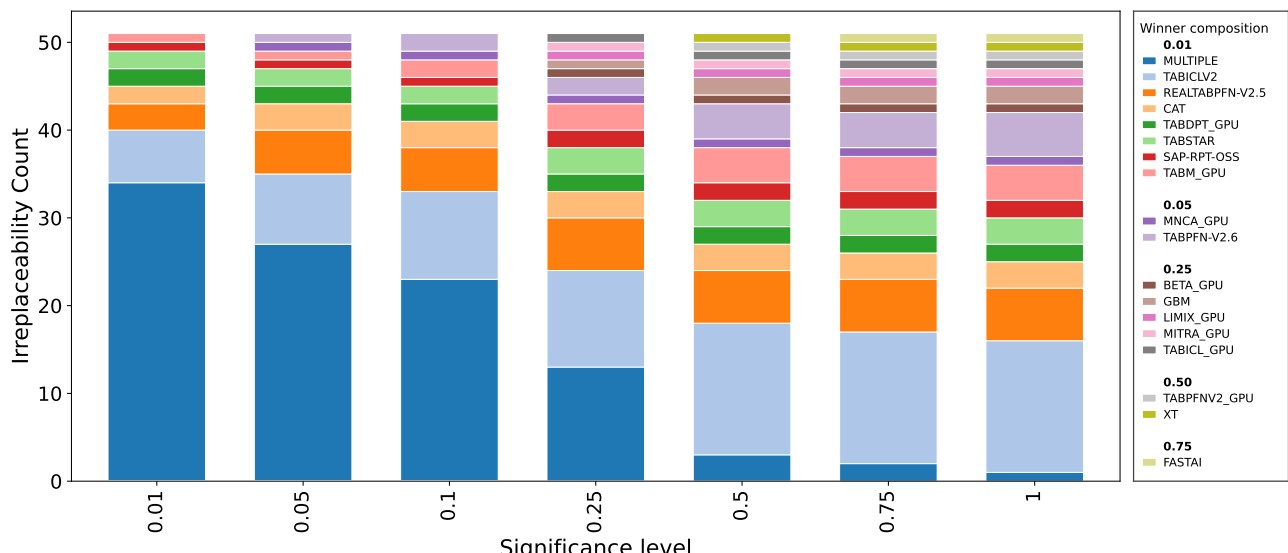

*Figure 5.* **Sensitivity analysis of irreplaceable models with different significance levels for the Wilcoxon signed-rank test.** The legend shows for each model starting from which significance level it is found to be irreplaceable for at least one dataset in the plot. A significance level of 1 stands for simple rank 1 counts without significance tests.

tions are irreplaceable for each model. It can be seen that all models benefit from tuning on the majority of datasets. This implies that, even though modern foundation models often perform well by default, they can only be compared adequately when the traditional alternatives are appropriately tuned.

**Irreplaceability of AutoML solutions**   To test the impact of multi-model ensembles as they are typically implemented in AutoML solutions, we add the AutoGluon results on the TabArena benchmark to our analysis. Figure 7 shows that for seven datasets, AutoGluon presets are irreplaceable. This indicates that AutoGluon, viewed as a model, has a favorable inductive bias incorporated in the pipeline that cannot be replaced by any of the benchmarked individual models.

**Historical view**   To account for the fact that new models are proposed over time and newer ones often build on top of findings in studies of previous models, we apply our irreplaceability metric to the TabArena benchmark in a sequential order. We start with the set of models that was already available before the increased interest in developing deep learning models for tabular data: tree-based models except of CatBoost, two simple MLP baselines, a linear model and kNN. We sequentially add each of the 20 newer TabArena models and analyze which of the models become a new irreplaceable model across the datasets in the benchmark. Note that the historical view that we adopt is not the exact publication order when a model was first proposed. Most of the methods received updates leading to the TabArena version that was benchmarked not being the initially proposed model version. Therefore, we track the model's GitHub release history if available, and the arXiv release history otherwise to construct an artificial "publication" order. The results can be seen in Figure 8. The presented view summarizes the progress of the whole field in a single figure and carries several pieces of information. The following patterns are particularly remarkable:

- Out of the initial model set, only CatBoost remains necessary to obtain peak performance on three datasets, while the capabilities of simple baselines and the other tree-based models can be replaced by newer models.
- On nine datasets, no new model improved over what was possible with previously available baselines.
- The most remarkable new addition was TabPFNv2, which at the time of publication would have become irreplaceable on 17 datasets, marking the first huge breakthrough achieved by foundation models.
- Four models have unique strengths on single datasets: TabPFN-2.6, SAP-RPT-OSS, ModernNCA, and TabDPT.
- Novel models are still able to make progress, also from the data-centric perspective, indicated by TabICLv2 being currently irreplaceable on 9 datasets.
- On many of the datasets where recent models made progress, other models already made progress prior to them. This might indicate that newer models slowly progress towards getting better at uncovering the true patterns behind the data generation process of those datasets.

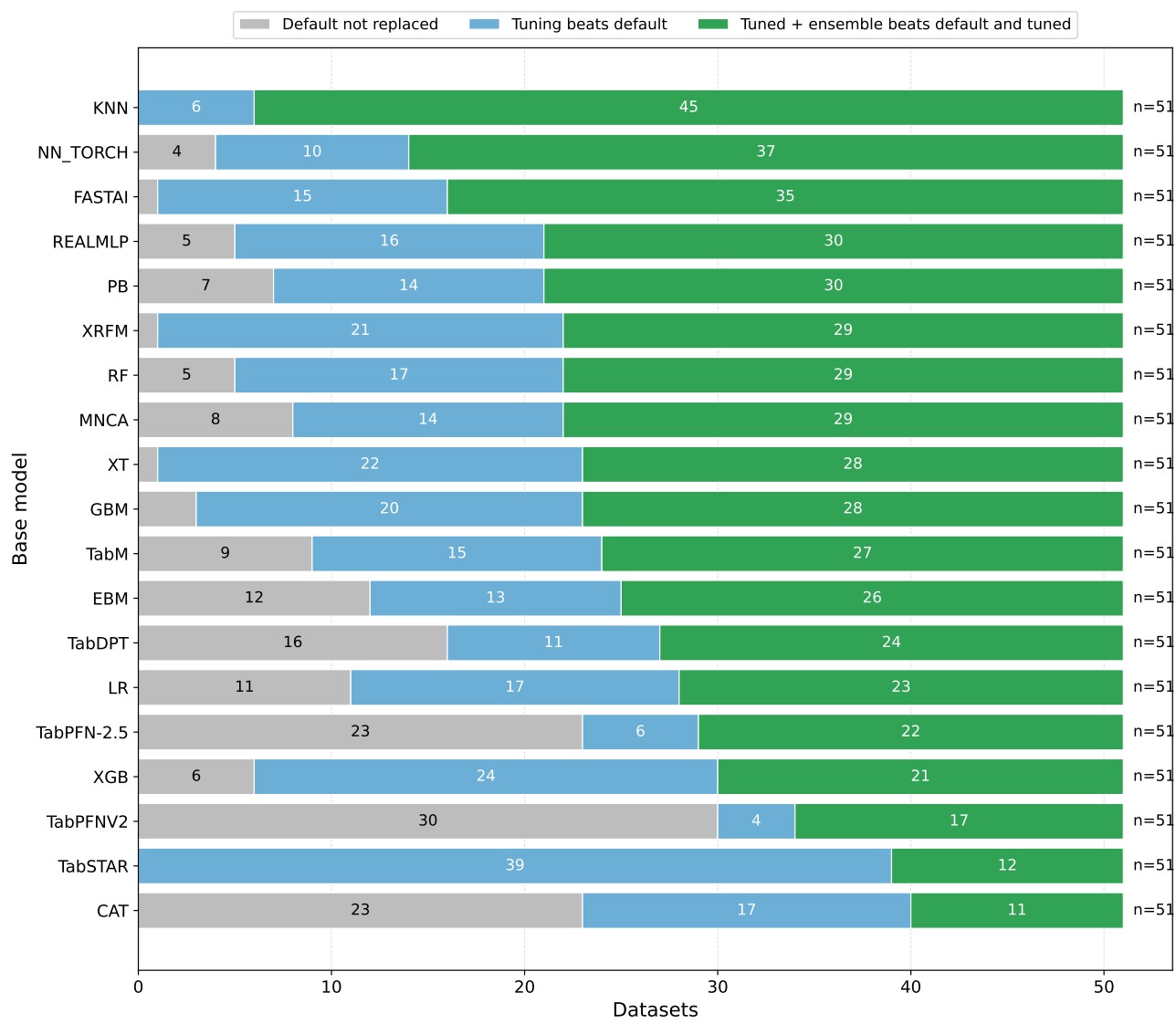

*Figure 6.* **Irreplaceability of hyperparameter optimization for each model.** Grey bars count on how many datasets the default model does neither significantly improve from tuning nor from post-hoc ensembling; blue how often tuning improves over default without post-hoc-ensembling improving over the tuned setting; and green how often post-hoc ensembling of configurations further improves over default as well as a single tuned configuration.

Lastly, to provide a compact view of Figure 4 accounting for the fact that some models were the first to reach certain performance levels, we add a further category to highlight this aspect in Figure 9.

## D. Limitations

Our framework is intended as a complementary view on benchmark performance, and since it may influence how future research evaluates model performance, we consider it important to discuss all limitations in detail, even if many of them are a matter of perspective rather than true limitations.

**Predictive performance is not the only objective.** We focus on peak predictive performance comparison, leaving all other aspects aside. In many applications, other factors such as training time, inference latency, memory use, hardware requirements, interpretability, robustness, calibration, or deployment constraints may be equally or even more important. A model that is irreplaceable in predictive performance may therefore still be unsuitable in practice.

*Figure 7.* **Combined view of the model performance dimensions with AutoGluon included.** We show the significant rank counts for each model focusing on the introduced metrics. 1st (sufficient) illustrates sufficiency minus the no. of datasets on which a model is irreplaceable to make the bar plots add up to 51 datasets. The models are sorted by ELO (rightmost is best).

**The frontier depends on the benchmark model zoo.** Irreplaceability and sufficiency are defined relative to the set of models included in the benchmark. If important competitors are missing, the estimated peak-performance frontier may be incomplete. Conversely, adding many closely related models can change the frontier and may reduce the apparent uniqueness of existing methods. The metrics are therefore most informative when the benchmark contains a sufficiently broad and well-maintained model zoo.

**Repeated evaluations are required.** Our statistical definitions rely on repeated measurements, such as repeated cross-validation or repeated train-test splits. In general, significance tests can be applied for single train/test split settings without repetition as well. However, without enough repeated observations, dataset-level significance tests are less reliable. Therefore, we view this as a desirable requirement for rigorous benchmarking, that should be applied even though it increases computational cost and may not always be feasible for very large models or datasets.

**Significance thresholds introduce discreteness.** The metrics depend on a chosen significance level $\alpha$. This introduces threshold effects: small changes in $\alpha$ can change whether a model is counted as sufficient, irreplaceable, redundant, or fallible. Nevertheless, explicit statistical thresholds are preferable to aggregation over p-values because they provide a more interpretable summary accounting for uncertainty with using thresholds based on conventions that stretch across various scientific communities. In practice, sensitivity analyses over multiple values of $\alpha$ should accompany the reported metrics.

**The metrics assume trustworthy evaluations.** Our framework inherits all limitations of the underlying benchmark. If the evaluation protocol is biased, the data splits are flawed, the implementations are unevenly optimized, or results are noisy, then the frontier metrics will reflect these issues. This is particularly important for strong labels such as irreplaceability, which may invite stronger conclusions than aggregate scores and should therefore be interpreted with caution.

**Model dependence can bias irreplaceability.** The framework implicitly treats benchmarked models as separate candidates. In practice, models may share components, training data, architectures, preprocessing pipelines, or implementation choices. This can bias irreplaceability, because a model may appear non-redundant or redundant partly due to the presence of closely related methods. One way to address this is to compute irreplaceability under different views of the model zoo, for example by excluding derivative models, grouping models into families, or using a historical evaluation that only compares against models available at a given time.

**The four categories are a simplification.** Our framework assigns each model-dataset pair to a small number of discrete categories. This is useful for interpretability, but it discards finer-grained information. In particular, the redundancy category can contain models that are close to the frontier as well as models that are only moderately competitive. Depending on the application, it may be useful to refine this category further, for example by distinguishing models that are statistically indistinguishable from the second-, third-, or lower-ranked method.

**Family imbalance may affect conclusions.** If one model family is much more heavily represented than others, then the chance of this family being irreplaceable or sufficient increases. In our setting, tabular foundation models are the most prominently represented family, which raises the possibility of multiple-comparison effects. A more conservative analysis

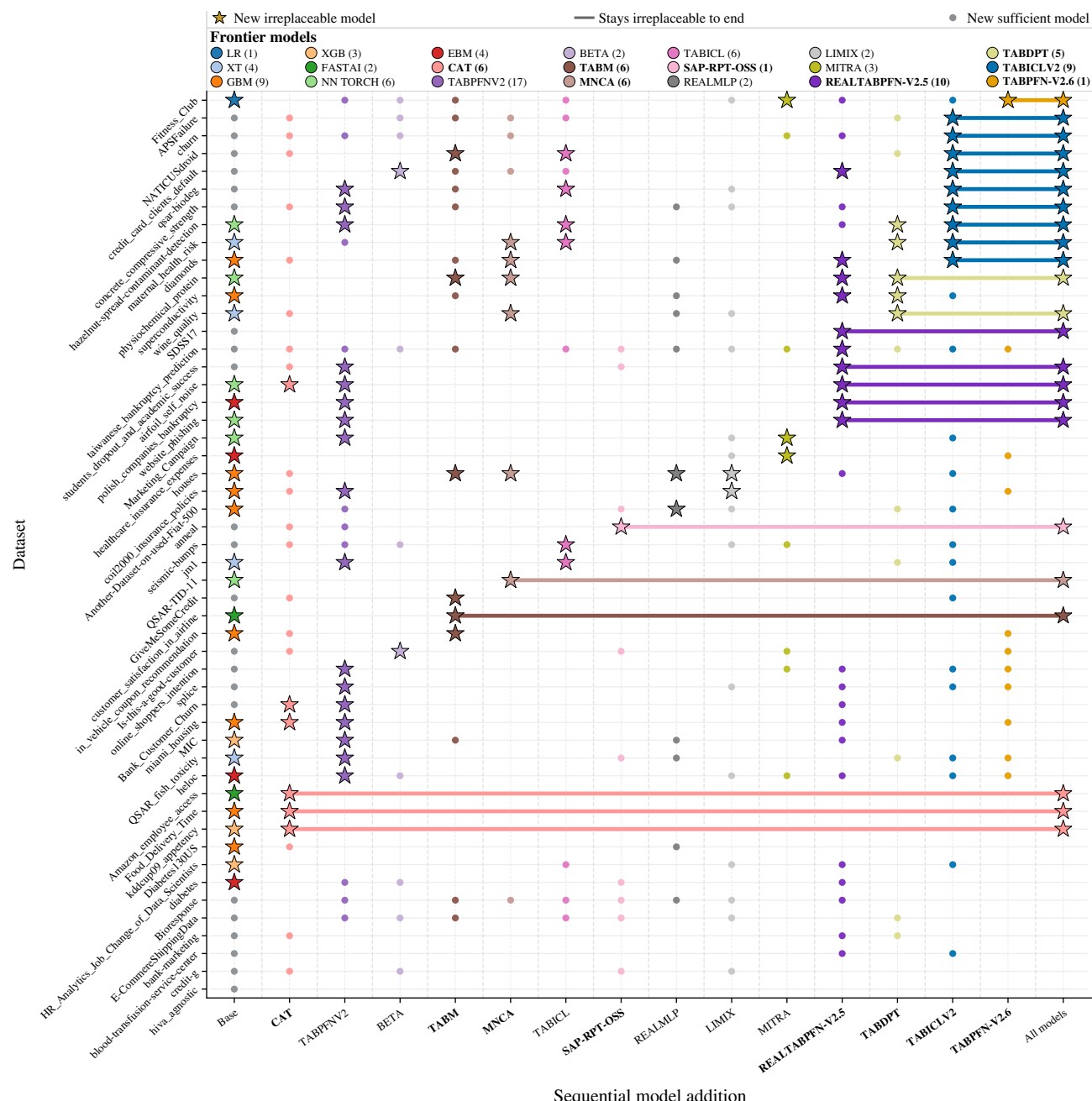

*Figure 8.* **Historical progression of how new models replaced older models establishing new peak performance.** Sequentially, for each dataset, each model on the x axis is compared to all previous models and is visualized with a star if it becomes irreplaceable, with a dot if it is sufficient, and not visualized otherwise. The order of the models is by the publication date of the model version that is evaluated in the TabArena benchmark. The last column shows a star for all models that are irreplaceable in the current benchmark. Models that were not irreplaceable on any dataset at publication are omitted from the plot. TabSTAR was omitted due to irreplaceability being misleading due to data leaks.

could control for multiple hypothesis testing or report additional views at the model-family level.

**Benchmarks are sequential, but the metric is static.** Our main definitions compare each model against all other models currently included in the benchmark. However, models are developed and added sequentially over time. A method that is no longer irreplaceable today may still have expanded the frontier when it was introduced. Historical or time-aware analyses are therefore useful for studying scientific progress, whereas the static view is better suited for assessing the current model zoo.

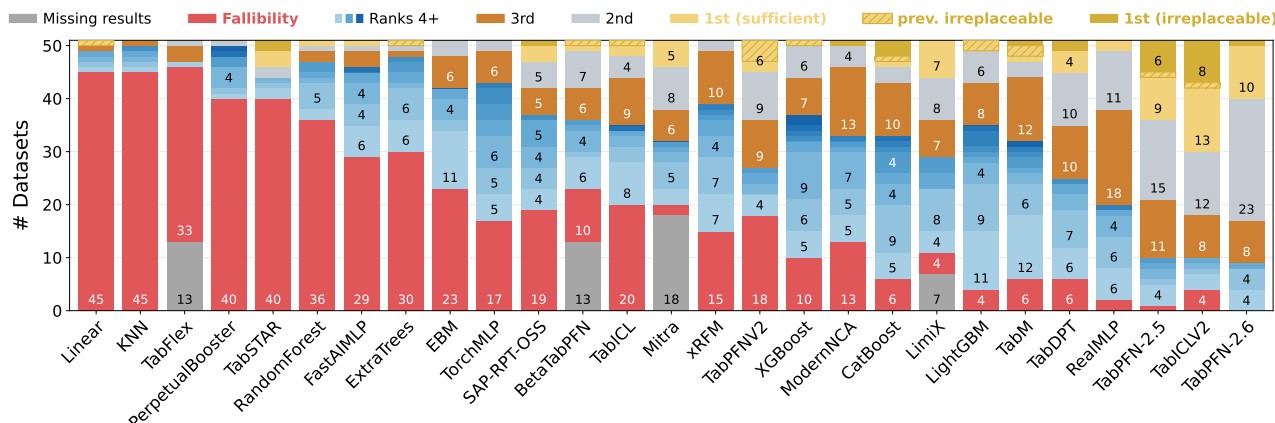

*Figure 9.* **The model performance dimensions with highlighting models that expanded the frontier and were the first to reach current peak performance.** We show the scores on the four metrics for each model. For sufficiency, we subtract the no. of datasets on which a model is irreplaceable to make the bar plots add up to 51 datasets.

