# OpenReview forum: "Lost in Aggregation: How Benchmarks Overlook Irreplaceable Model Strengths"
_ICML.cc/2026/Workshop/FMSD — FMSD @ ICML 2026 Poster_

### Official Review · Reviewer_YwVQ · 2026-05-21
**A Well-Executed Study of Benchmarking Beyond Performance Aggregation**

**Rating:** 7
**Confidence:** 4

**Review:**

### **Summary**

The paper argues that standard aggregate benchmark scores, such as average ranks, Elo scores, or win rates, are useful for identifying robust default models but can hide dataset-specific model strengths. To address this, the authors propose peak-performance frontier metrics for tabular benchmarks. These metrics classify models according to whether they are *irreplaceable*, *sufficient*, *redundant*, or *fallible* on each dataset. Applied to TabArena, the analysis shows that some models with unique dataset-specific strengths can appear weaker under aggregate metrics, while recent tabular foundation models tend to expand the peak-performance frontier.

### **Strengths**

The paper is very well written and easy to follow. The proposed peak-performance frontier metrics are sound, clearly motivated, and address a real limitation of aggregate benchmark reporting.

The analysis on TabArena is insightful. In particular, it shows that aggregate metrics often measure consistent decent performance, while underemphasizing dataset-level peak performance. Models with similar aggregate performance can excel on different datasets.

Several ablation and complementary studies strengthen the contribution. These include the historical progression analysis, the study of how new models establish new peak performance while replacing older models, and the analysis of how hyperparameter tuning and ensembling affect irreplaceability.

The paper also includes a thoughtful limitations section, acknowledging that predictive performance is not the only relevant deployment criterion and that the metrics depend on the benchmark model zoo.

Overall, the empirical evaluation is solid, and the paper is highly relevant to the workshop.

### **Weakness/Areas for Improvement**

Overall, the paper’s idea is interesting and well executed. My main concern is that the practical utility of the proposed metrics could be better articulated.

Practitioners would typically not rely only on aggregate benchmark metrics when selecting a model for deployment. Instead, they would benchmark several models on their own use case, which already implicitly accounts for whether a model is uniquely strong for that specific dataset. This somewhat limits the practical novelty of the proposed framework.

Relatedly, it would be useful to clarify what added value is obtained from knowing that a model is irreplaceable on some number of benchmark datasets if it is not the best overall default model. The paper could further discuss how sensitive the conclusions are to the chosen model zoo. Since irreplaceability is relative to the set of included models, adding or removing closely related models may change which methods appear uniquely valuable.

The practical adoption of the proposed metrics could be better justified by explaining how they should be used alongside aggregate metrics. For example, a recommended workflow could be: use aggregate metrics to identify strong default candidates, then use irreplaceability and sufficiency to identify complementary models worth including in a model-selection.

While the proposed metrics offer a richer interpretation of benchmark results, it appears that models performing well under the proposed criteria (particularly irreplaceability and sufficiency) often also rank highly under aggregate benchmark metrics. If so, the practical incremental value of the framework for model selection is less clear. The authors could better articulate when these metrics would lead to meaningfully different conclusions than standard aggregate rankings, and whether their main value is diagnostic interpretation rather than changing model selection decisions.

Another aspect that could be improved is the analysis of inductive biases. The paper mentions that irreplaceability may reveal dataset-specific strengths, but it would be valuable to analyze what is common across datasets where a given model is irreplaceable. For example, are these datasets characterized by small sample size, categorical-heavy features, high dimensionality, strong interactions, or noisy labels? This would make the results more actionable for model development.

A minor presentation issue is that some figures, such as Figures 6, 8, and 9, appear not to be exported in vector/PDF format, making them hard to read. The font size of several figures could also be increased.

### **Detailed Comments**

The authors should include a more explicit discussion of how practitioners should use these metrics in practice. Are they mainly intended for benchmark designers, model developers, or end users selecting models?

The paper would be stronger with a dataset-level characterization of irreplaceability. For each model with nonzero irreplaceability, it would be useful to summarize the properties of the datasets where it uniquely reaches the frontier.

The authors should clarify whether irreplaceability should be interpreted as evidence of a model’s unique inductive bias or simply as a benchmark-dependent observation.

The figures should be improved for readability, especially in the appendix.

### **Justification of Score**

This is a clear accept for me. The paper is well written, well motivated, and proposes a useful complementary perspective on tabular model benchmarking. The empirical analysis on TabArena is convincing and relevant to the workshop.

---

### Official Review · Reviewer_Qtj6 · 2026-05-21
**The paper introduces a framework to disentangle the average performance of tabular tasks to distinguish utility-based predictors from dataset-specific predictors.**

**Rating:** 7
**Confidence:** 4

**Review:**

## Strenghts

- The paper describes an important tension when predictors are evaluated: their average performance can obscure specific strengths of some methods.
- The performance frontier described by the irrepleacible, sufficient, redundant, and fallible categories is novel and helps to detect in practice which model works better for a particular dataset.
- The conclusions are well supported by the empirical evidence of the TabArena benchmark.

## Areas for Improvement

- The main argument seems to be a classical one in statistics: the expected value (or aggregate performance in this case) is not a sufficient descriptor of the entire distribution.  Yet little analysis is conducted on the distribution of errors. So a question naturally arises: is the framework correlated to some measure of the dispersion of the errors?
- How does this look in practice? In production environments, this is usually the process followed: a bunch of methods are compared, and then the best one is selected. How does this framework improve current practice?

## Detailed comments

- The perspective on the relevant unit of analysis being the dataset is important since aggregates can obscure specific behavior. But this assertion carries a deeper assumption: what is a dataset?  One could argue that the unit of analysis is also artificial, in that its construction implies prior assumptions, and that performance in a dataset is also an aggregate. So, iterating on this argument, perhaps the relevant unit of analysis is the observation. This should be addressed in the paper.

---

### Official Review · Reviewer_vmnP · 2026-05-22
**Lost in Aggregation: How Benchmarks Overlook Irreplaceable Model Strengths**

**Rating:** 7
**Confidence:** 4

**Review:**

The paper argues that standard benchmark aggregation metrics (Elo, average rank, win rate, and normalized error) primarily reward consistency and failure avoidance, and systematically obscure whether a model is uniquely necessary to achieve peak performance on any individual dataset. The authors introduce a four-category framework: irreplaceable, sufficient, redundant, and fallible, defined relative to a statistically grounded peak-performance frontier, and apply it to TabArena's 27 models across 51 datasets. The central finding is that standard metrics are nearly perfectly correlated with each other, but largely uncorrelated with irreplaceability, meaning models with unique dataset-specific strengths can rank poorly on every standard leaderboard.

# Strengths

- I found the core insight of the paper to be underappreciated. It's an important observation that the authors are tackling. A model ranked 6th on every dataset can rank 1st under Elo, and the average rank is a clean proof-of-concept. RealMLP being #1 on TabArena at publication, never irreplaceable, being a frontier on only 2 datasets makes this concrete and memorable.
- The framework is statistically rigorous. Paired Wilcoxon signed-rank tests on repeated cross-validation folds to define the frontier, rather than raw point estimates, correctly account for evaluation uncertainty at the dataset level.

# Areas of improvement
- I thought that the framework's conclusions are sensitive to the choice of $\alpha$. Figure 5 shows that model irreplaceability changes substantially between α=0.01 and α=0.25. The paper recommends sensitivity analyses but doesn't offer a principled default or discuss what α values are appropriate for different use cases.
- The appendix proposes a staged evaluation strategy (low fallibility → high sufficiency → irreplaceable models) but doesn't quantify the expected gain from following it versus just using the Elo leaderboard directly. Without this, the framework is a diagnostic tool without providing a clear decision framework.

# Recommendation

Accept for workshop. The central argument is correct, well-evidenced, and timely for a community heavily invested in benchmark rankings. The framework is immediately applicable to TabArena and transferable to other benchmarks. The limitations are well disclosed.